# Immunostimulatory Activity of Lactic Acid Bacteria Cell-Free Supernatants through the Activation of NF-κB and MAPK Signaling Pathways in RAW 264.7 Cells

**DOI:** 10.3390/microorganisms10112247

**Published:** 2022-11-13

**Authors:** Jaekoo Lee, Seonyoung Kim, Chang-Ho Kang

**Affiliations:** MEDIOGEN Co., Ltd., Biovalley 1-ro, Jecheon-si 27159, Korea

**Keywords:** lactic acid bacteria, RAW264.7 macrophage cells, immune-stimulating activity, probiotics

## Abstract

Lactic acid bacteria (LAB) can improve host health and has strong potential for use as a health functional food. Specific strains of LAB have been reported to exert immunostimulatory effects. The primary goal of this study was to evaluate the immunostimulatory activities of novel LAB strains isolated from humans and foods and to investigate the probiotic properties of these strains. Cell-free supernatants (CFS) obtained from selected LAB strains significantly increased phagocytosis and level of nitric oxide (NO) and pro-inflammatory cytokines such as tumor necrosis factor (TNF)-α and interleukin (IL)-6 in RAW264.7 macrophage cells. The protein expression of inducible NO synthase (iNOS) and cyclooxygenase (COX)-2, which are immunomodulators, was also upregulated by CFS treatment. CFS markedly induced the phosphorylation of nuclear factor-κB (NF-κB) and MAPKs (ERK, JNK, and p38). In addition, the safety of the LAB strains used in this study was demonstrated by hemolysis and antibiotic resistance tests. Their stability was confirmed under simulated gastrointestinal conditions. Taken together, these results indicate that the LAB strains selected in this study could be useful as probiotic candidates with immune-stimulating activity.

## 1. Introduction

The human immune system plays a critical role in protecting the body from bacteria, viruses, and harmful substances, and comprises innate and adaptive immunity [1,2]. In general, innate immunity is considered a non-specific response, whereas the adaptive immune system plays a role in inducing an antigen-specific immune response.

Innate immunity involves immune cells such as dendritic cells, macrophages, neutrophils, eosinophils, and natural killer (NK) cells. Macrophages play a crucial role in removing dead cells, tumor cells, and foreign compounds through phagocytosis induction [3]. In addition, macrophages have been reported to affect adaptive immunity by secreting proinflammatory cytokines such as interferon (IFN)-γ, tumor necrosis factor (TNF)-α, and interleukin (IL)-1β, IL-6, and IL-12, and producing many mediators such as inducible nitric oxide synthase (iNOS), cyclooxygenase-*2* (COX-2), nitric oxide (NO), and prostaglandin E2 (PGE2) [4,5]. NO synthesized by iNOS is involved in immune regulation, inflammation, tumor destruction, and non-specific host defense [6]. TNF-α and IL-6 are among the first substances produced during the innate immune response to bacteria and are associated with the activation of macrophages [7]. The synthesis of these mediators is mainly induced by activation of the nuclear factor-κB (NF-κB) and mitogen-activated protein kinase (MAPK) signaling pathways [5].

NF-κB signaling pathway, which is one of the most studied signaling pathways in inflammatory responses, regulates the immune response by modulating gene expression involved in immune and inflammatory responses [7]. In particular, NF-κB is involved in the activation of macrophages via the induction of cytokine production [8]. MAPK signaling pathways are also related to macrophage activation. Numerous studies have reported that MAPK signaling pathways are increased in RAW264.7 cells by various stimuli [9]. ERK and p38 regulate gene expression associated with innate immunity, and JNK can control the expression of iNOS [10].

Probiotics are characterized by living microorganisms that benefit the host’s health when consumed in moderate amounts [11]. Probiotics, such as lactic acid bacteria (LAB) and *bifidobacteria*, have been reported to improve intestinal health, enhance immunity, and reduce serum cholesterol [11,12]. Recently, as interest in applying probiotics to improve immunity has increased, research on the immunomodulatory effects of probiotics has been actively conducted. Some strains of LAB including *Lactobacilli*, *Lactococcus*, *Bifidobacterium* have been found to have an immunostimulatory effect on macrophage cells [13,14,15].

The present study evaluated the immunostimulatory activity and associated molecular mechanisms of LAB, including *Limosilactobacillus reuteri* (*L. reuteri*), *Limosilactobacillus fermentum* (*L. fermentum*), and *Lactococcus lactis* (*Lc. lactis*) isolated from humans, fermented foods, and food in RAW 264.7 macrophage cells. In addition, we investigated the probiotic properties, such as safety and gastrointestinal tract (GIT) tolerance, for their potential use as probiotics.

## 2. Materials and Methods

### 2.1. LAB Strains and Cultivation Conditions

All probiotic strains tested in this study were obtained from MEDIOGEN Co., Ltd. (Jecheon, South Korea). All strains were cultivated and maintained in MRS broth (Difco Laboratories, Detroit, MI, USA) at 37 °C under anaerobic conditions. The origins and species of all the LAB strains used in the experiments are shown in Table 1.

### 2.2. Preparation of Cell-Free Supernatants (CFS)

To evaluate the immunostimulatory activity of the strains used in this study, CFS was prepared by using a slightly modified version of the previously published method [16]. Briefly, the LAB strains were cultured in MRS broth at 37 °C for 18 h. Then, CFS was obtained from bacterial cultures by centrifugation at 4000× *g* for 20 min, followed by filtration using 0.2 μm filters, and then freeze-dried. The freeze-dried CFS was resuspended in DMEM and used for the cell culture studies.

### 2.3. Cell Culture

RAW 264.7 cells were obtained from the Korea Cell Line Bank (Seoul, South Korea). RAW 264.7 cells were maintained in Dulbecco’s modified Eagle’s medium (DMEM; Welgene Inc., Gyeongsan, South Korea) supplemented with 10% heat-inactivated fetal bovine serum and 1% penicillin–streptomycin (Welgene Inc.). Cells were routinely cultivated at 37 °C in a humidified atmosphere containing 5% CO_2_. The cells were sub-cultured at 70–80% confluence.

### 2.4. Cell Viability Assay

The cytotoxicity of RAW264.7 cells was investigated using the 3-[4,5-Dimethylthiazole-2-yl]-2,5-diphenyltetrazolium bromide (MTT) assay (Sigma-Aldrich Inc., St. Louis, MO, USA) according to a previous study [4]. Briefly, RAW 264.7 cells were seeded into 96-well plates at a density of 1 × 10^5^ cells/well (100 μL/well) and cultivated at 37 °C. When the RAW 264.7 cells reached 70% confluence, CFS was added to each well, and the plates were incubated at 37 °C for 24 h. Thereafter, RAW 264.7 cells were washed three times with phosphate-buffered saline (PBS), MTT solution (100 μL) was added to each well, and the cells were further incubated for 4 h. After removing the MTT solution, the formed MTT formazan was dissolved in 100 μL dimethyl sulfoxide (DMSO). Absorbance was measured at 570 nm using an Epoch 2 microplate reader (Biotek Instruments Inc., Winooski, VT, USA).

### 2.5. Determination of NO Production

NO released from RAW 264.7 cells was measured by the Griess assay according to a previous study [4]. Briefly, RAW 264.7 cells were seeded in 96-well plates at a density of 1 × 10^5^ cells/well and then cultivated overnight. The cells were stimulated with CFS (5 mg/mL) for 24 h. Lipopolysaccharide (LPS; 10 ng/mL; Sigma-Aldrich Inc.) was used as the positive control. The collected culture supernatants were mixed with Griess reagent (1% sulfanilamide and 0.1% N-(1-naphthyl)-ethylenediamine dihydrochloride) at room temperature for 15 min. Absorbance was measured at 540 nm using a microplate reader.

### 2.6. Neutral Red Uptake Assay for Phagocytic Activity

The phagocytic activity of RAW 264.7 cells was investigated using a neutral red uptake assay, as previously described [5]. Briefly, RAW 264.7 cells (1 × 10^5^ cells/well) were added to 96-well plates and cultivated at 37 °C with 5% CO_2_. After overnight growth, the cells were treated with CFS (5 mg/mL) for 24 h. Thereafter, RAW 264.7 cells were washed three times with PBS, 100 μL of neutral red solution (0.01%) was added to each well, and the cells were incubated for 2 h. After washing three times, each well was treated with a cell lysis solution (acetic acid: ethanol = 1:1), and the plate was incubated at 37 °C for 1 h. Absorbance was measured at 540 nm using a microplate reader.

### 2.7. Determination of Cytokines Production

Cytokine analysis was performed according to the previous study [17]. Briefly, RAW 264.7 cells were seeded in 24-well plates at a density of 2 × 10^5^ cells/well and incubated overnight. The cells were then treated with fresh DMEM containing LPS (10 ng/mL) or CFS (5 mg/mL) for 24 h. The cytokines (TNF-α and IL-6) released into the cell culture medium were measured using ELISA kits (R&D Systems, Minneapolis, MN, USA) according to the manufacturer’s guidelines.

### 2.8. Western Blot Analysis

Western blot analysis was conducted according to the previous study with some modifications [18]. After 24 h of treatment with LPS (10 ng/mL) or CFS (5 mg/mL), RAW 264.7 cells were lysed with RIPA buffer (Gendepot Inc., Katy, TX, USA) containing phosphatase inhibitor cocktails at 4 °C for 20 min. The total protein concentration in the cell lysates was measured using Bradford assay (Gendepot Inc.). Proteins (40 μg) were separated using 10% SDS-PAGE and transferred onto a polyvinylidene fluoride membrane (Millipore, Billerica, MA, USA) for 40 min. After transfer, the membrane was blocked at room temperature for 1 h with 5% BSA in 0.1% PBS-T and probed with specific primary antibodies (iNOS, COX-2, ERK, p-ERK, JNK, p-JNK, p38, p-p38, NF-κB-p65, p-NF-κB-p65, and β-actin) at 4 °C overnight. The PVDF membrane was then washed three times with 0.1% PBS-T and incubated with horseradish peroxidase-conjugated secondary antibodies in a blocking buffer at room temperature for 1 h. After additional washing, protein bands were developed using the ECL system. β-actin was used as a loading control. 

### 2.9. Probiotic Properties

#### 2.9.1. Resistance to Simulated Gastrointestinal Conditions

The gastrointestinal resistance of selected LAB strains was investigated as previously described by Qi, et al., [19]. Bacterial cells were cultured for 18 h at 37 °C in MRS broth and collected by centrifugation (4000× *g* for 20 min). Pellets were washed twice with PBS and resuspended in PBS. The optical density of the cell suspension solution was adjusted to 1.0 at 600 nm. One milliliter of cell suspension was added to 9 mL of simulated gastric fluid (SGF; 0.3% pepsin in PBS, pH 2.5) and cultivated at 37 °C for 2 h. The cell pellet was obtained by centrifugation (4000× *g* for 20 min), mixed with 10 mL of stimulated intestinal fluid (SIF; 1% pancreatin and 1% bile salt in PBS, pH 7.4), and then cultivated at 37 °C for 2.5 h. Viable cells were determined by plate counting on MRS agar and expressed as log CFU mL^−1^. The survival rate was calculated using the following Equation (1):Survival rate (%) = (Log N/Log N_0_) × 100(1)
where Log N is the log number of viable cells at the end of the test and Log N_0_ is the log number of initial viable cells.

#### 2.9.2. Hemolytic Activity

To assess the hemolytic activity, the selected LAB strains were cultivated overnight in MRS broth. Bacterial cultures were streaked onto tryptic soy agar (Difco Laboratories) plates containing 5% sheep blood (MB cell, Seoul, Korea) and cultivated for 48 h at 37 °C. Hemolytic activity was classified based on the lysis of red blood cells around the colonies and was determined as α-hemolysis (greenish zone), β-hemolysis (transparent zone), or γ-hemolysis (no notable zone) [20].

#### 2.9.3. Antibiotic Susceptibility

Antibiotic susceptibility of the selected LAB strains was assessed using the antibiotic minimum inhibitory concentration (MIC) strip method according to the manufacturer’s guidelines (bioMérieux, Marcy-l’Étoile, France). The strains were cultured in MRS broth at 37 °C. After incubation for 18 h, the bacterial cells were obtained from bacterial cultures by centrifugation at 4000× *g* for 20 min, washed twice with PBS, and resuspended in PBS to a McFarland standard of 0.5. The cell suspension was spread over plates of brain infusion agar (Difco, Detroit, MI, USA) using a cotton swab, and the plates were allowed to dry for 10 min at room temperature. MIC test strips containing antimicrobial agents were placed on the agar surface. The plates were cultivated for 24 h at 37 °C, and cut-off values were determined according to the European Food Safety Authority (EFSA) guidelines [21].

### 2.10. Statistical Analysis

All data are expressed as the mean ± standard deviation (SD) from triplicate experiments. Differences between groups were evaluated for statistical significance using one-way analysis of variance (ANOVA), followed by Duncan’s pair-wise comparisons at *p* < 0.05 (SPSS, version 21; IBM Inc., Armonk, NY, USA).

## 3. Results

### 3.1. The Effect of CFS from LAB Strains on Cell Viability and NO Production in RAW 264.7 Cells

To evaluate whether CFS from LAB strains could increase NO production in cells, CFS was added to RAW 264.7 cells and incubated for 24 h. CFS from LAB strains had no cytotoxic effect on RAW 264.7 cells (Table 1). In addition, treatment with CFS from the LAB strains significantly increased NO production by 2.84–25.73 μM. In particular, *L. reuteri* MG5462 (21.73 ± 0.95 μM), *L. fermentum* MG4263 (21.47 ± 1.14 μM), MG4268 (20.87 ± 2.02 μM), and MG4282 (25.73 ± 0.12 μM) showed similar or higher induction compared to LPS (19.27 ± 2.80 μM).

Since NO is associated with the function of macrophages, such as the eradicating of cell debris, pathogens, and tumor cells, increased NO production can activate the immune response in macrophages [22]. Therefore, *L. reuteri* MG5462, *L. fermentum* MG4263, MG4268, MG4282, and *Lc. lactis* MG4668, MG5278, and MG5474, which have high NO production, were selected, and further studies were conducted with these strains.

### 3.2. Effect of CFS from LAB Strains on Phagocytic Activity in RAW 264.7 Cells

The phagocytic activity of CFS from the LAB strains was assessed using a neutral red uptake assay. Cellular morphology and statistical data analysis showed that phagocytic activity was significantly enhanced by more than 2-fold in CFS-treated RAW 264.7 cells compared to untreated control cells (Figure 1a,b). This result was similar to that obtained for the LPS-treated positive control.

### 3.3. Effect of CFS from LAB Strains on iNOS/COX-2 Expression in RAW 264.7 Cells

The effect of CFS from LAB strains on iNOS and COX-2 protein expression in RAW 264.7 cells was analyzed using a Western blot assay. We found that cells treated with CFS of all strains, except that of *Lc. lactis* strain MG4668, significantly induced iNOS expression compared with that in the untreated control group (Figure 2a,c). In particular, the expression levels of iNOS in cells treated with *L. reuteri* MG5462 and *L. fermentum* MG4263, MG4268, and MG4282 increased by approximately 6-fold compared to untreated control cells and by approximately 2-fold compared to LPS-treated cells, which was consistent with the results of NO production.

In addition, the expression levels of COX-2 in all CFS-treated cells ranged from 13- to 23-fold compared to untreated control cells, and CFS-treated cells of all strains, except MG4282, were more than 2-fold higher than LPS-treated cells (Figure 2b,d). These results suggest that CFS from LAB strains may induce proinflammatory mediators, such as iNOS and COX-2, by activating cells.

### 3.4. Effect of CFS from LAB Strains on Cytokine Production in RAW 264.7 Cells

To evaluate the immune-enhancing activity of CFS from LAB strains, we investigated the concentrations of TNF-α and IL-6 secreted by RAW264.7 cells. The levels of cytokines released into the cell culture medium were determined using an ELISA kit. TNF-α and IL-6 levels in LPS-stimulated RAW 264.7 cells (72.67 ± 3.30 and 4.81 ± 0.81 ng/mL, respectively) were significantly elevated compared to those in untreated cells (Figure 3a,b). In addition, TNF-α levels were significantly higher in all CFS-treated cells, except for *L. fermentum* MG4282, than in LPS-treated cells. In particular, *L. reuteri* MG5462 (176.94 ± 12.15 ng/mL) showed the highest TNF-α level, approximately 2-fold higher than that in LPS-treated cells (72.67 ± 3.30 ng/mL).

The IL-6 concentrations of all CFS-treated cells were remarkably elevated compared with those of the untreated control cells. In particular, *L. fermentum* MG4282 (17.85 ± 1.06 ng/mL) showed the highest IL-6 production, approximately 3-fold compared to LPS-treated cells (4.81 ± 0.81 ng/mL), followed by *L. fermentum* MG4263 (9.94 ± 2.81 ng/mL), *Lc. lactis* MG5278 (9.92 ± 1.42 ng/mL), and *Lc. lactis* MG5474 (8.45 ± 1.87 ng/mL).

### 3.5. Effect of CFS from LAB Strains on NF-κB and MAPKs Activation in RAW 264.7 Cells

MAPKs such as ERK, JNK, p38, and NF-κB are representative signaling pathways associated with cell activation [9]. In this study, we assessed whether CFS induces the activation of NF-κB and MAPKs using Western blotting to understand the signaling pathways modulating the immune-enhancing activity of CFS from LAB strains. Our results showed that CFS from all strains significantly promoted the phosphorylation of NF-κB in RAW 264.7 cells compared to that in untreated control cells (Figure 4).

In addition, phosphorylation of ERK, JNK, and p38 was induced by all CFS treatments and LPS compared to that in the untreated control group (Figure 5).

### 3.6. Probiotic Properties of the LAB Strains

#### 3.6.1. Stability of LAB Strains under Simulated Gastrointestinal Conditions

The stability of the LAB strains was investigated by incubating the strains in artificial GIT for a fixed period (Table 2). After 4.5 h of exposure to simulated gastrointestinal conditions, the survival rate of the selected strains ranged from 31.24% to 97.41%. In particular, *L. reuteri* MG5462 (97.41 ± 0.02%) showed the highest survival rate.

#### 3.6.2. Safety Evaluation of LAB Strains as Probiotics

To confirm whether LAB are safe for use as probiotics, we investigated their hemolytic activity using a blood agar test. As shown in Figure 6, all selected LAB strains exhibited white colonies after 48 h of incubation on blood agar plates, suggesting that LAB strains have no hemolytic activity.

In addition, MIC tests were performed to evaluate the antibiotic susceptibility of LAB strains (Table 3). All other strains, except *L. reuteri* MG5462 and *L. fermentum* MG4263, were susceptible to the selected antibiotics compared to the microbiological breakpoint values defined by the EFSA guidelines. *L. reuteri* MG5462 and *L. fermentum* MG4263 were resistant to chloramphenicol.

## 4. Discussion

Because of their immunomodulatory activities, probiotics can be applied as a strategy to prevent intestinal diseases, including diarrhea and irritable bowel syndrome, and extraintestinal diseases, such as allergies [23]. In particular, probiotics interact with dendritic cells, T cells, and B lymphocytes to regulate the innate and adaptive immunity of the host [24,25]. Recently, numerous studies have reported that probiotics can stimulate the immune system by producing cytokines [14,15,26,27]. Therefore, this study aimed to identify functional probiotic candidate strains to improve the immune system through immunostimulatory activity.

The innate immune system, which plays a leading role in the body’s first line of defense and protection, helps destroy pathogens and foreign substances from entering the body. Numerous immune cells such as dendritic cells, macrophages, and NK cells are involved in the innate immune response. In particular, these cells play key roles in the inflammatory response and immune enhancement. In terms of immune enhancement, macrophages have been reported to play a critical role in the host defense system through phagocytosis [28].

Lately, much interest has been paid to CFS, which contains bioactive soluble factors secreted by probiotics [29,30]. The CFS of probiotic strains was used as an in vitro model to assess their physiological benefits [30]. In this study, we focused on the effect of CFS from LAB strains on the immunostimulatory activity of macrophages. Phagocytosis by macrophages is an essential process for removing pathogens and dead cells [31]. In previous studies, CFS of *Lacticaseibacillus rhamnosus* (*L. rhamnosus*) GG effectively enhanced the phagocytic activity of murine cells J774 [32]. Xiu, et al., [33] reported that exopolysaccharides derived from *Lacticaseibacillus casei* (*L. casei*) WXD030 enhanced the phagocytic activity in RAW264.7 cell. Similarly, in our study, CFS from LAB strains increased the phagocytic activity of cells (Figure 1).

Activated cells stimulate the production of various immunomodulators such as NO, iNOS, COX-2, and proinflammatory cytokines [5,34]. NO is a key biomolecule that mediates a wide range of biological functions, including immunomodulation, inflammation, tumor suppression, and defense against pathogen infection [6]. iNOS and COX-2, which promote the production of NO and PGE2, respectively, activate the immune response in stimulated cells [35]. Furthermore, proinflammatory cytokines, such as TNF-α, IL-6, and IL-1β, secreted by activated cells are known to be involved in the destruction of pathogens and tumor cells [36]. Therefore, these immunomodulators can produce immunostimulatory effects in innate immune responses. According to previous studies, the culture supernatant of *Lactiplantibacillus plantarum* (*L. plantarum*) B0040, *L. plantarum* B0110, and *Weissella cibaria* (*W. cibaria*) B0145 increased the production of NO, TNF-α and IL-6 in RAW 264.7 cells [17]. Ren, et al., [37] reported that CFS from different LAB strains (*L. plantarum*, *L. casei*, *L. reuteri*, *L. fermentum*, *Lactobacillus acidophilus*, *L. rhamnosus*, and *Levilactobacillus brevis*) significantly induced TNF-α secretion from THP1 cells. Furthermore, the viable *W. cibaria* JW15 and *L. rhamnosus* GG significantly increased the protein expression of iNOS and COX-2 in murine macrophages [27]. With the same results, our study showed that CFS treatment significantly enhanced the production of NO, TNF-α, and IL-6 in RAW 264.7 cells and upregulated iNOS and COX-2 protein expression. These results suggest that the selected LAB strains may induce immune enhancement by activating macrophages.

NF-κB, which stimulates the production of many cytokines, is involved in regulating innate and adaptive immune functions [38]. Phosphorylation of p65, an NF-κB subunit, promotes NF-κB activation [39]. Certain strains of LAB have shown immunostimulatory activity by inducing the phosphorylation of p65 in cell models [40,41]. In the present study, we confirmed that CFS significantly induced phosphorylation of NF-κB p65 (Figure 4). This result suggests that the selected LAB strains induced the phosphorylation of p65 to stimulate NF-κB activation.

MAPK signaling pathways play key roles in cell proliferation, differentiation, and activation [42]. MAPK phosphorylation induces the transcriptional activation of NF-κB [43]. According to the previous study, bioactive soluble factors secreted by *Leuconostoc lactis* induced the phosphorylation of MAPKs in RAW264.7 cells [44]. In this study, we identified that CFS effectively stimulates the phosphorylation of MAPKs such as ERK, JNK, and p38 in RAW264.7 cells (Figure 5). These results suggest that CFS of the selected LAB strains increased the activation of the NF-κB and MAPK signaling pathways, consequently inducing the production of immunomodulators such as NO, TNF-α, and IL-6.

Ingested probiotics are exposed to unfavorable conditions such as gastric fluid and bile salts [45]. Thus, probiotic strains must survive in the human gastrointestinal tract and colonize the colon to confer health benefits to the host [46]. Some strains of LAB including *Lactobacilli* and *Lactococcus* have high survival rates under gastrointestinal conditions [47,48]. However, in these studies, bacteria were incubated in culture medium or peptone water to help them survive. In this study, we evaluated the viability of LAB strains in artificial GIT that did not contain any compounds to protect probiotic strains and confirmed that the selected LAB strains exhibited a 31.2–97.4% survival rate in simulated GIT (Table 2). According to a previous study, the viability (%) of specific strains in the simulated GIT was similar to the results of the current study [49]. Therefore, these findings suggest that all the selected LAB strains have the potential to be used as probiotics. *L. fermentum*, *L. reuteri*, and *Lc. lactis* are generally considered as safe (GRAS) [50,51,52]. However, the safety of probiotics for use in food products should be carefully recommended [53]. Thus, hemolysis and antibiotic susceptibility were determined to confirm the safety of probiotics. In the present study, we found that all selected LAB strains exhibited a γ-hemolytic effect on blood agar plates (Figure 6). However, *L. reuteri* MG5462 and *L. fermentum* MG4263 were resistant to chloramphenicol (Table 3). Pathogens can acquire antibiotic resistance genes via lateral gene transfer from probiotics with antibiotic resistance [54]. Intrinsic resistance caused by mutations in chromosomal genes reduces the risk of horizontal transfer; therefore, further studies on potentially mobile elements such as transposons and plasmids are required [55].

## 5. Conclusions

In conclusion, our study demonstrated that CFS of *L. reuteri* MG5462; *Lc. lactis* MG4668, MG5278, and MG5474; and *L. fermentum* MG4263, MG4268, and MG4282 induce immunostimulatory activity in RAW264.7 cells. In the current study, treatment with CFS from LAB strains stimulated the phagocytosis of macrophage cells and enhanced the expression of immunomodulators such as NO, TNF-α, IL-6, iNOS, and COX-2 by activating the NF-κB and MAPK signaling pathways. Further studies are required to determine the efficacy of these strains in an in vivo cyclophosphamide-induced immunosuppression model. However, our results suggest that the selected LAB strains are potential probiotic candidates with immunopotentiation effects.

## Figures and Tables

**Figure 1 microorganisms-10-02247-f001:**
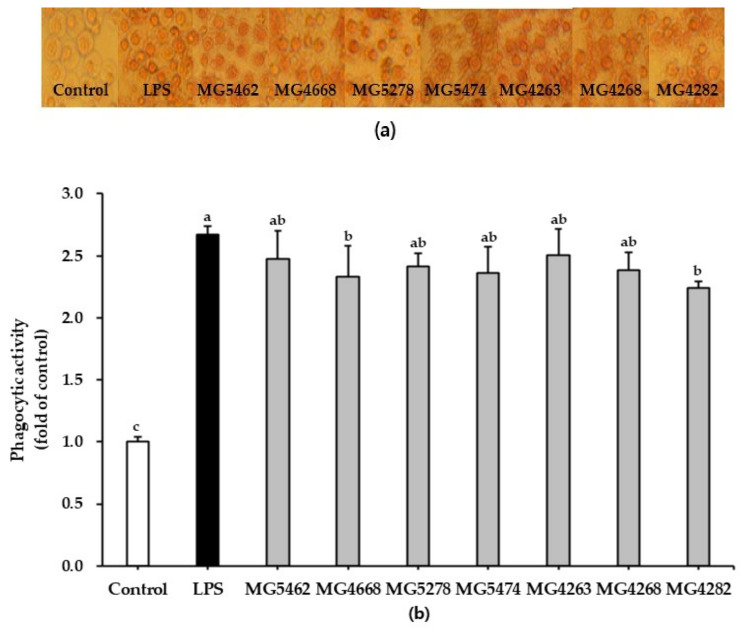
Effect of CFS from LAB strains on the phagocytosis of RAW264.7 macrophage cells. Cells were treated with LPS (10 ng/mL) or CFS (5 mg/mL) for 24 h, and phagocytosis was evaluated by adding neutral red dye for 2 h. (**a**) The morphology of the untreated (control) or treated cells was visualized by phase-contrast light microscopy at a 20× magnification. (**b**) Phagocytic activity was quantified by measuring the logarithmic intensity using a microplate reader. Data are presented as the mean ± SD of three independent experiments (*n* = 3). Different letters indicate significant differences between means at *p* < 0.05 based on Duncan’s multiple range test.

**Figure 2 microorganisms-10-02247-f002:**
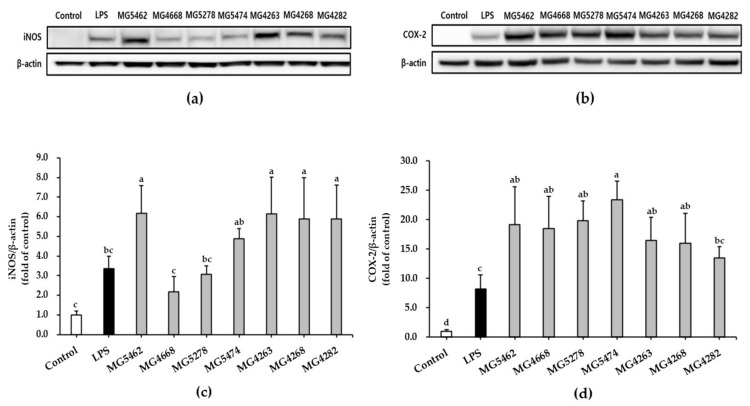
Effect of CFS from LAB strains on the protein expression levels of iNOS and COX-2 in RAW264.7 cells. Representative Western blot and expression of iNOS (**a**,**c**) and COX-2 (**b**,**d**). Cells were treated with LPS (10 ng/mL) or CFS (5 mg/mL) for 24 h. The protein expression levels were quantified by densitometry, and the data were normalized relative to the intensity of β-actin. Untreated cells (control) were treated with DMEM only. Data are presented as the mean ± SD of three independent experiments (*n* = 3). Different letters indicate significant differences between means at *p* < 0.05 based on Duncan’s multiple range test.

**Figure 3 microorganisms-10-02247-f003:**
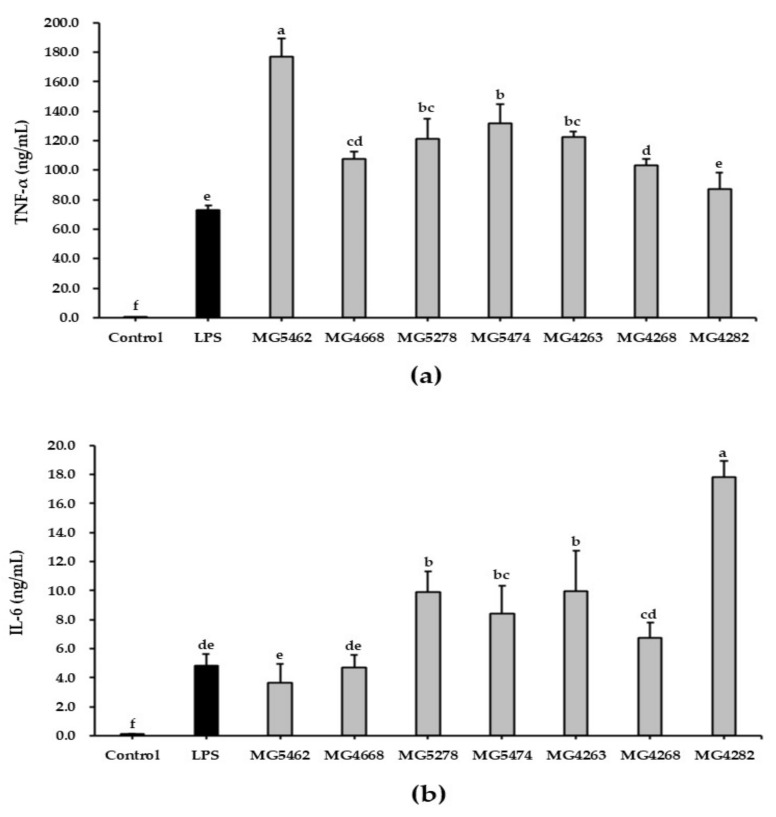
Effect of CFS from LAB strains on cytokine production in RAW264.7 cells. Cells were treated with LPS (10 ng/mL) or CFS (5 mg/mL) for 24 h. The amounts of TNF-α (**a**) and IL-6 (**b**) secreted by cells were measured using an ELISA kit. Data are presented as the mean ± SD of three independent experiments (*n* = 3). Different letters indicate significant differences between means at *p* < 0.05 based on Duncan’s multiple range test.

**Figure 4 microorganisms-10-02247-f004:**
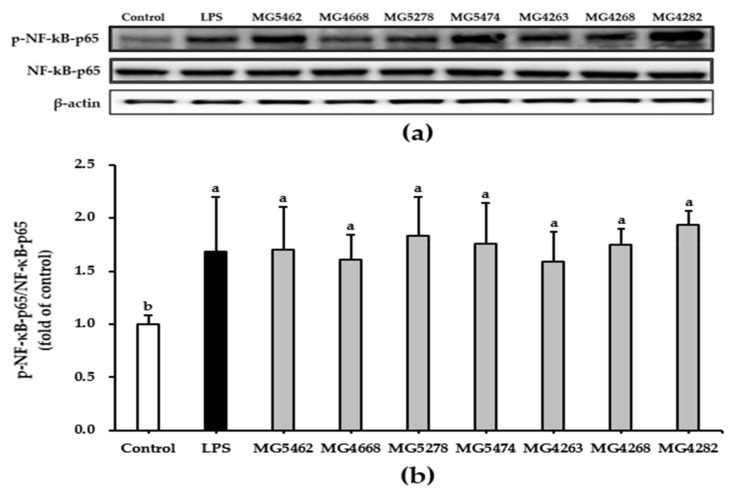
Effect of CFS from LAB strains on NF-κB activation in RAW264.7 cells. Representative Western blot (**a**) and activation levels of NF-κB (**b**). Cells were treated with LPS (10 ng/mL) or CFS (5 mg/mL) for 24 h. The NF-κB activation levels were quantified using densitometry. Data are presented as the mean ± SD of three independent experiments (*n* = 3). Different letters indicate significant differences between means at *p* < 0.05 based on Duncan’s multiple range test.

**Figure 5 microorganisms-10-02247-f005:**
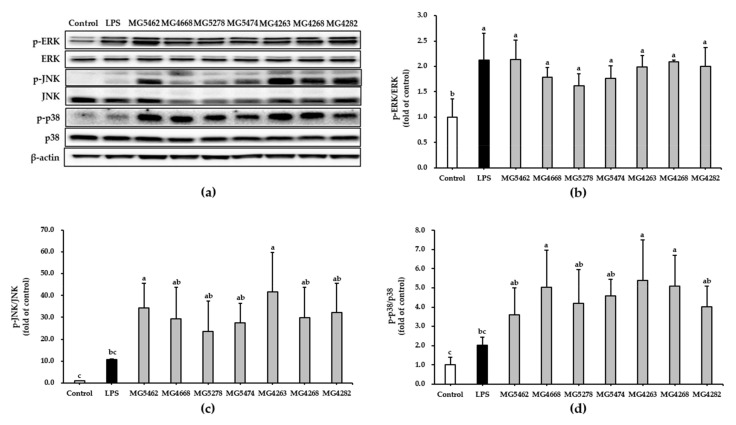
Effect of CFS from LAB strains on MAPK activation in RAW264.7 cells. Cells were treated with LPS (10 ng/mL) or CFS (5 mg/mL) for 24 h. Representative Western blot (**a**) and the activation levels of MAPK; ERK (**b**), JNK (**c**), and p38 (**d**) were quantified by densitometry. Data are presented as the mean ± SD of three independent experiments (*n* = 3). Different letters indicate significant differences between means at *p* < 0.05 based on Duncan’s multiple range test.

**Figure 6 microorganisms-10-02247-f006:**
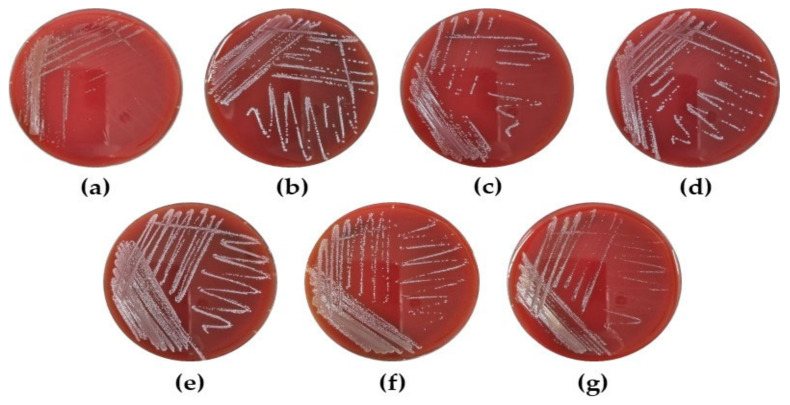
Hemolysis of LAB strains. (**a**) *L. reuteri* MG5462, (**b**) *Lc. lactis* MG4668, (**c**) *Lc. lactis* MG5278, (**d**) *Lc. lactis* MG5474, (**e**) *L. fermentum* MG4263, (**f**) *L. fermentum* MG4268, and (**g**) *L. fermentum* MG4282.

**Table 1 microorganisms-10-02247-t001:** Effect of CFS from LAB strains on cell viability and NO production in RAW264.7 cells.

Origin	Strains	Cell Viability (%)	NO (μM)
	Untreated	100.00 ± 0.48 ^a^	1.00 ± 0.06 ^h^
	LPS (10 ng/mL)	99.69 ± 2.92 ^a^	19.27 ± 2.80 ^c^
Food	*Lacticaseibacillus rhamnosus* MG5007	101.84 ± 3.40 ^a^	4.11 ± 0.40 ^g^
	*Lactococcuslactis* MG5399	98.92 ± 5.21 ^a^	2.84 ± 0.29 ^g^
	*Lactococcuslactis* MG5403	100.56 ± 0.77 ^a^	4.09 ± 1.09 ^g^
	*Limosilactobacillus reuteri* MG5462	101.25 ± 1.96 ^a^	21.73 ± 0.95 ^b^
	*Lactococcuslactis* MG5479	103.88 ± 1.47 ^a^	6.94 ± 0.52 ^f^
	*Lactococcuslactis* MG5508	100.49 ± 2.17 ^a^	3.23 ± 0.44 ^g^
Fermented food	*Lacticaseibacillus paracasei* MG5179	100.88 ± 4.82 ^a^	4.81 ± 0.21 ^g^
	*Lactococcuslactis* MG5126	101.08 ± 1.74 ^a^	4.28 ± 1.04 ^g^
	*Lactococcuslactis* MG5278	100.62 ± 0.98 ^a^	10.60 ± 0.72 ^e^
	*Lactococcuslactis* MG5452	100.95 ± 1.04 ^a^	3.13 ± 0.76 ^g^
	*Lactococcuslactis* MG5474	101.40 ± 0.76 ^a^	14.13 ± 0.90 ^d^
	*Lactococcuslactis* MG5542	100.05 ± 2.67 ^a^	3.99 ± 1.32 ^g^
Human	*Limosilactobacillus fermentum* MG4263	100.65 ± 2.48 ^a^	21.47 ± 1.14 ^b^
	*Limosilactobacillus fermentum* MG4268	101.29 ± 3.44 ^a^	20.87 ± 2.02 ^bc^
	*Limosilactobacillus fermentum* MG4282	101.26 ± 1.77 ^a^	25.73 ± 0.12 ^a^
	*Bifidobacterium lactis* MG4568	100.05 ± 2.67 ^a^	4.69 ± 0.10 ^g^
	*Bifidobacterium lactis* MG4569	102.83 ± 1.08 ^a^	4.73 ± 0.16 ^g^
	*Lactococcuslactis* MG4668	101.51 ± 1.65 ^a^	8.04 ± 1.55 ^f^
	*Limosilactobacillus reuteri* MG4722	101.58 ± 2.02 ^a^	4.17 ± 0.03 ^g^
	*Limosilactobacillus reuteri* MG5149	100.09 ± 3.12 ^a^	4.58 ± 0.27 ^g^

Data are presented as the mean ± SD of three independent experiments (*n* = 3). Different letters indicate significant differences between the means at *p* < 0.05, based on Duncan’s multiple range test. CFS: cell-free supernatant; LAB: Lactic acid bacteria; NO: nitric oxide.

**Table 2 microorganisms-10-02247-t002:** Survival rate of the LAB strains under simulated gastrointestinal conditions.

Strains	Initial Counts (log CFU/mL)	SGF (log CFU/mL)	SIF (log CFU/mL)	Survival Rate (%)
*L. reuteri* MG5462	7.70 ± 0.01	7.66 ± 0.00	7.50 ± 0.00	97.41 ± 0.02
*Lc. lactis* MG4668	7.59 ± 0.06	2.85 ± 0.00	2.75 ± 0.10	36.27 ± 1.27
*Lc. lactis* MG5278	7.29 ± 0.06	2.74 ± 0.04	2.61 ± 0.01	35.78 ± 0.07
*Lc. lactis* MG5474	7.66 ± 0.02	2.78 ± 0.07	2.39 ± 0.36	31.24 ± 4.66
*L. fermentum* MG4263	7.87 ± 0.01	3.27 ± 0.01	3.09 ± 0.01	39.25 ± 0.09
*L. fermentum* MG4268	7.89 ± 0.02	4.11 ± 0.05	3.74 ± 0.15	47.40 ± 1.84
*L. fermentum* MG4282	7.92 ± 0.01	3.96 ± 0.03	3.92 ± 0.08	49.58 ± 0.98

Data are presented as the mean ± SD of three independent experiments (*n* = 3). LAB: Lactic acid bacteria; SGF: simulated gastric fluid; SIF: stimulated intestinal fluid.

**Table 3 microorganisms-10-02247-t003:** Antibiotic susceptibility of LAB strains.

Antibiotics	Susceptibility (S/R)						
	MG5462	MG4668	MG5278	MG5474	MG4263	MG4268	MG4282
Ampicillin	S	S	S	S	S	S	S
Gentamicin	S	S	S	S	S	S	S
Kanamycin	S	S	S	S	S	S	S
Streptomycin	S	S	S	S	S	S	S
Tetracycline	S	S	S	S	S	S	S
Chloramphenicol	R	S	S	S	R	S	S
Erythromycin	S	S	S	S	S	S	S
Vancomycin	n.r	S	S	S	n.r	n.r	n.r
Clindamycin	S	S	S	S	S	S	S

Antibiotic susceptibility was determined according to EFSA guidelines [21]. The antibiotic susceptibility of LAB strains is classified as susceptible (S) or resistant (R) according to the established cut-off values from EFSA. Data are presented as mean ± SD of three independent experiments (*n* = 3). LAB: lactic acid bacteria; n.r.: not required.

## Data Availability

The data presented in this study are available on request from the corresponding author.

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
