# Peer review of "Immunostimulatory Activity of Lactic Acid Bacteria Cell-Free Supernatants through the Activation of NF-κB and MAPK Signaling Pathways in RAW 264.7 Cells"

_microorganisms, 2022, doi:10.3390/microorganisms10112247_

Round 1
Reviewer 1 Report
Comments to the Author
MANUSCRIPT DETAILS
Ms. Ref. No.: microorganisms-2009796
Title: Immune-stimulating activity of lactic acid bacteria through the activation of NF-κB and MAPK signaling pathways in RAW 264.7 cells
Article Type: Article
JOURNAL: Microorganisms
GENERAL COMMENTS
The objective of this study is to evaluate the immunostimulatory activities of novel LAB strains isolated from humans and foods and to investigate the probiotic properties of strains’ cell-free supernatants (CFS)
The interest in this manuscript is significant, the MS is well designed and discussed.
However, the MS needs a moderate language polishing.
The comments and questions provided below may help the authors to put their MS in enhanced form.
Comments:
1. To be more precise; as the MS discussing Immune-stimulating activity of lactic acid bacteria cell-free supernatants (CFS) on cell line, I suggest to refer to this point in the MS title to be; “Immune-stimulating activity of lactic acid bacteria cell-free supernatants through the activation of NF-κB and MAPK signaling pathways in RAW 264.7 cells”
2. Kindly revise to provide the missed references to all methods illustrated in the MS e.g. Preparation of Cell-free supernatants (CFS), Determination of Cytokines production.. Revise thoroughly.
3. L338, 347: Some strains of LAB including Lactobacillus sp. Lactobacillus and Bifidobacterium. As authors should focus on discussion of tested strains. Kindly, revise thoroughly for similar comments.
4. The language needs polishing e.g. L292: “The innate immune system, which plays a leading role in the body’s first line of defense ,and protection. helps destroy pathogens and foreign substances from entering the body. Kindly, revise thoroughly for similar comments.
5. L358: that selected LAB strains’ CFS induce.. Kindly, revise thoroughly for similar comments.
Author Response
"Please see the attachment."

Reviewer 2 Report
This paper titled “Immune-stimulating activity of lactic acid bacteria through the activation of NF-κB and MAPK signaling pathways in RAW 264.7 cells” is interesting. Thus, I think this manuscript could be considered for publication in Microorganisms after Major revising.
My comments are as follow:
1. Check the abbreviations, the text format of full text.
2. The introduction section is too simple.
3. Discussion section: please compared your results to previous studies.
4. English should be improved.
Author Response
"Please see the attachment."

Reviewer 3 Report
Dear Authors
Your article needs minor revision, especially in the introduction and conclusion. the materials and methods are well presented, but the results and discussion need more explanation and cite more recent articles.
Author Response
"Please see the attachment."

Reviewer 4 Report
In this research article, the authors studied the immunomodulatory potential of LAB isolated from various sources on the macrophage cell line RAW 264.7, in vitro. They found that cell-free supernatants (CFS) increased NO production, without affecting cell viability. Selected strains also induced phagocytotic activity and upregulated the production and secretion of pro-inflammatory cytokines. Lastly, the authors showed that CFS activated the NF-κB and MAPK signaling pathways. The conclusions of this work are supported by the experimental data provided; however, some points require attention:
1. The time of incubation should be added in section 2.8. Phosphorylation of proteins is an event that happens very rapidly after treatments, and thus more information about the protocol followed should be added here.
2. The nomenclature of strains used in the study should follow the guidelines proposed by Zheng et al., 2020 (https://pubmed.ncbi.nlm.nih.gov/32293557/). Also, species names should be italicized.
3. The rationale for selecting 7 strains out of 20 is not clear in the text, a comment should be added.
4. Figure 2 is missing.
5. Please be careful when making claims about probiotics. To the best of my knowledge, no probiotic supplement has been cleared for therapeutic use in gastrointestinal or extraintestinal diseases (Lines 41-43, 283-285). Appropriate corrections should be made in the text.
Author Response
"Please see the attachment."

Reviewer 5 Report
This study is very well written in terms of the introduction and discussion and the results is very well presented too. The inclusivity of various testing is also commended. However, I see a clear limitation here which is the use of LPS as a control which makes it difficult to judge whether the immunostimulatory effects which is well-observed (and very clear) by the potential probiotic strains could be considered a positive or negative immune modulation. LPS is a known pro-inflammatory and that was stated in this study as well. There was no justification for the use of LPS; perhaps a pathogen control would be another idea. So, unless LPS use is justified and a clear discussion about how the immuno-stimulation observed would be considered a positive aspect, particularly that most of the results of the lactic acid bacterial CFS are comparable to the LPS, I would find it hardly convincing that these immunostimulatory properties would be a positive outcome.
Author Response
"Please see the attachment."

Round 2
Reviewer 2 Report
Authors addressed all my comments.
Reviewer 5 Report
Happy with the authors' response